# Towards the Evaluation of Augmented Reality in the Metaverse: Information Presentation Modes

**Michele Gattullo** , **Enricoandrea Laviola** *, **Alessandro Evangelista** , **Michele Fiorentino**  and **Antonio Emmanuele Uva**

Department of Mechanics, Mathematics, and Management, Polytechnic Institute of Bari, Via Orabona, 4, 70125 Bari, Italy
* Correspondence: enricoandrea.laviola@poliba.it

**Abstract:** In the future, many activities will be carried out in the Metaverse: hybrid offices and video-based education are just some examples. The way research is carried out could change, too. In this context, this work investigates the possibility of simulating Augmented Reality (AR) user studies on information presentation in a virtual environment. Organizing an industrial setup is complex; thus, most studies are executed in laboratories. However, lab experiments present limitations, e.g., the number and variety of participants and the availability of facilities. User studies may also be carried out by exploiting simulated AR, as an initial step for the Metaverse, where people are connected regardless of their location. This alternative could be used to carry out experiments on AR information presentation to solve common issues, such as the lack of physical equipment to perform component location tasks and the long time required to collect a large sample of users. Indeed, researchers could propose CAD models with information that simulates the same visual realism achieved with true AR. Moreover, multiple tests could be conducted in parallel by not relying on a limited amount of physical equipment per user. In this work, we developed and evaluated a desktop-simulated testing environment (DSTE) to conduct AR information presentation experiments remotely. We applied it in a pilot user study, revealing that the proposed DSTE was effective for the related research goals. Furthermore, 40 participants reported a positive user experience. The evaluation confirms that using a DSTE is promising for collecting and analyzing data from a wide range of people.

**Keywords:** AR simulation; user interaction; information presentation; user study; industrial metaverse

## 1. Introduction

The effectiveness of Augmented Reality (AR) for information presentation in industrial applications is widely asserted in the literature [1–5]. Various AR system prototypes have been developed with different interface approaches to identify the optimal way to present information. User studies are a fundamental method to find an optimal interface design and define usable guidelines. However, the proportion of user study papers among all AR papers is still low [6]. Limitations due to SARS-CoV-2 restricted the possibility of carrying out experiments in the presence of others and forced researchers to explore new solutions for AR user studies. It is reasonable that in the future, user studies will be carried out in the Metaverse [7–9] instead of laboratories. This work represents an initial step towards this scenario. In fact, we developed and evaluated a desktop-simulated testing environment (DSTE) that allows users to carry out experiments on information presentation in AR without being physically present in a laboratory.

In the industrial field, the validation of AR applications is crucial since they must deal with complex requirements [10] involving various factors, such as people, technical aspects, regulations, environments, and profits. Organizing user studies in the natural industrial setup is challenging; thus, most studies are executed in laboratory environments [6]. However, lab-based studies can involve only users who can physically access



laboratories. For example, it is difficult to recruit industrial operators or involve people from different geographical locations. Thus, the number and variety of participants in the sample are actually limited, and, therefore, the effects of social factors such as language, culture, and education are hard to demonstrate. Furthermore, in some cases, laboratories are not accessible due to ongoing renovation, a lack of laboratory staff, and safety reasons, as experienced during the SARS-CoV-2 epidemiological emergency.

A solution to increase the number of people involved in user studies is to simulate real-world content through a virtual environment. The virtualization of the testing environment is coherent with the emerging concept of the Metaverse, a non-face-to-face service that breaks the boundaries between real-world space and virtual space [7,9]. It is generally intended as an immersive, three-dimensional, virtual, and multi-user environment that allows people to interact with each other regardless of their location using computational tools such as simulations [8]. In this way, experiments may be carried out from a remote location (e.g., the user's office or house) rather than in a real testing environment. This technique, which is called simulated AR [11], indirect AR [12], or immersive virtual AR [13], was effectively used as a design tool before, both to simplify the test execution and to increase the number of people involved in the user study. However, only in recent years has it become more and more widespread as a precursor to the Metaverse, especially after the SARS-CoV-2 pandemic. In addition, previous works [13–16] used complex Virtual Reality (VR) devices that are not commonly owned by users and are hard to distribute.

Thus, the main novel aspect of our research is the development of a testing tool for desktop devices that exploits simulated AR. The aim is to allow users to carry out experiments on information presentation in AR regardless of their location and use their personal computer/laptop with a mouse and keyboard. In particular, this tool could overcome common issues, such as the lack of physical equipment to perform component location tasks and the long time required to collect a large sample of users. The proposed DSTE reproduces a LEGO assembly experiment, a common testing scenario in the literature on this topic [17–20]. We have adopted some expedients, in terms of interaction and visual realism, to make the simulated AR scenario as close as possible to the true one. In this way, users may better perceive that not all visualized elements are effectively designed to be virtual, unlike common desktop applications that are entirely in VR. In this work, we wanted to address the following research questions:

1. Can the proposed DSTE replace true AR in formal user studies to evaluate information presentation?
2. How is the user experience with the proposed DSTE?
3. How is the user satisfaction with the proposed interaction with simulated real objects compared to the real ones in true AR?

To answer these research questions, we first evaluated the DSTE by applying it in a pilot user study to optimize the use of visual assets [21] in AR. We asked participants to assess their user experience and satisfaction with the DSTE through a survey based on the User Experience Questionnaire (UEQ) [22]. Then, at the end of the pilot experiment, we also gathered feedback from experimenters through a focus group.

In Section 2, we present the related work about the use of simulated AR for user studies. In Section 3, we describe the DSTE. In Section 4, we present the results of the user evaluation of the DSTE, which we discuss in Section 5. In Section 6, we provide a conclusion.

## 2. Related Work

AR is an emerging technology that is becoming increasingly important in the industrial field [23,24] thanks to its capability of showing virtual information referenced in a real context. Although its potentialities are widespread, there are still many challenges for AR usage in industrial scenarios, such as hardware constraints, environmental changeability, interaction methods, and tracking issues [25–27]. These challenges lead to AR applications developed with poor versatility and a long development cycle [28]. Moreover, as stated by

Marques et al. [29], the literature is lacking in studies on how visual information should be correctly delivered. It is necessary to understand which visual assets are understood and accepted by users before their usage in a real scenario. In this regard, therefore, it is necessary to find a solution that ensures rapid experiments to evaluate the behavior of a large number of users in order to obtain consistent data. To achieve this goal, in our work, we propose a testing tool for desktop devices that exploits simulated AR to carry out experiments on AR information presentation.

In recent years, the design of AR interfaces was supported by simulating real environments through virtual environments (VEs). VEs have been used in numerous different domains, such as vehicle design, architecture, construction, and industrial plant design [30]. Previous works [12–16,31] revealed that using a simulated AR system for user evaluation ensures results that are comparable to true AR, but with lower costs and without the problem of managing different devices and real-life locations. As pointed out by Alce et al. [13], simulated AR applications should have two main characteristics to be effective. First, they should offer an adequate degree of virtual immersion, intended as the capability to deliver "an inclusive, extensive, surrounding, and vivid illusion of reality [32]". Second, the 3D interaction with the virtual environment (navigation and the manipulation of objects) should be intuitive. However, while the first characteristic can be managed by providing the desired level of visual realism [31], the user's 3D interaction in simulated AR is completely different from true AR [33] and highly depends on the display used. Users navigate and manipulate virtual objects rather than real ones with all of the additional technical issues related to physical interactions (e.g., the unpredictability of virtual object behavior, collision, and attaching) [11]. Alce et al. [13] proposed a solution to overcome the problem of grabbing unpredictability in a simulated VE using a virtual wristband that changes color and gives users vibration feedback when they grab the correct virtual object. Moreover, in order to facilitate immersion, ease of interaction, and physical awareness, they equipped the simulated VE with a virtual representation of users' own bodies. A similar approach was used by Lee et al. [14] with a simulated real hand. In their work, the collision issue was also discussed and solved by using high-contrast colors for interacting objects. Wang et al. [34] developed a system that estimated the hand pose to directly manipulate a virtual model during assembly task planning by calculating the hand strain. In these and other similar works [11,15,16], 3D interaction is then accomplished using complex devices, such as Virtual Reality Head-Mounted Displays, game controllers, motion capture gloves, and depth cameras. However, these devices are not easily distributable to users to carry out experiments from a remote location.

Considering the benefits of simulated AR [12–16,31], in this work, we propose a DSTE that allows users to carry out experiments remotely on information presentation in AR. This solution could be used to solve common issues, such as the lack of physical equipment to perform component location tasks and the long time required to collect a large sample of users. Indeed, researchers could propose CAD models with information that simulates the same visual realism achieved with true AR, as stated in [31]. Moreover, multiple tests could be conducted in parallel by not relying on a limited amount of physical equipment per user. Our proposal is focused only on supporting the choice of visual assets to convey information in AR. Especially for this purpose, the critical issues found in the literature [13] can be easily overcome. Then, an interaction with a partial degree of immersion could be also accepted. Therefore, we propose the use of simulated AR in a desktop testing environment that can be run on a user's personal computer. We used intuitive 3D interaction metaphors inspired by other user interfaces that exploit a mouse and keyboard, such as desktop Virtual Reality testing applications [35] and videogame Graphical User Interfaces (GUIs).

## 3. The Desktop-Simulated Testing Environment (DSTE)

We designed the DSTE for a target desktop device, equipped with a mouse and keyboard, independent of its screen resolution. In the proposed DSTE, users have to assemble

virtual LEGO Duplo bricks to create abstract shapes based on instructions provided through visual assets [36,37] (e.g., CAD models, drawings, and videos). It is possible then to evaluate user performance and/or preference with different experimental conditions regarding information presentation in AR.

In the following paragraphs, we discuss how we managed the visual realism and the 3D interaction issues in the DSTE, providing further implementation details. First, our design principle aims to offer the possibility of carrying out experiments from remote locations with conditions different from those of laboratories (e.g., the physical absence of experimenters for control and explanations and external disturbances). Then, we simplified the user experience with the DSTE, considering that difficult interaction with the VE would bias prototype evaluations, as Alce et al. [10] argued. In this way, it is possible to have reliable data even if users are not supervised during the experiments. However, if possible, we recommend recording or sharing a live screen for more accurate control and gathering information about errors.

### 3.1. Visual Realism in the DSTE

We decided to keep the DSTE as essential as possible without using a photorealistic rendering. Virtual reproductions of objects not relevant to the experiment were not included, such as picking bins and workplace facilities. The virtual objects that simulate their corresponding real ones were then limited to the LEGO bricks and a 26 × 26 green LEGO Duplo plate on which users place them. Even though the DSTE does not have a high degree of fidelity with a real testing environment, it does not invalidate the results of the experiments. We were careful that there was no altering of the information provided through visual assets or how they combine with the simulated real objects. Virtual LEGO bricks that simulate real ones are rendered in "opaque" mode, whereas CAD models used as visual assets are rendered in "transparent" mode with a semi-transparent shade (alpha 150 in the range from 0 to 255), as made in [17,38,39] (see Figure 1a). We took from [36] visual assets that can be tested with the proposed DSTE, and they can be either world-fixed (see Figure 2①) or screen-fixed (see Figure 2②). Screen-fixed visual assets are placed in the top-right corner of the interface. We also paid attention to the world-fixed ones to reproduce the effect of registration error through a slight misalignment between the simulated real bricks and the corresponding visual assets (see Figure 1b).

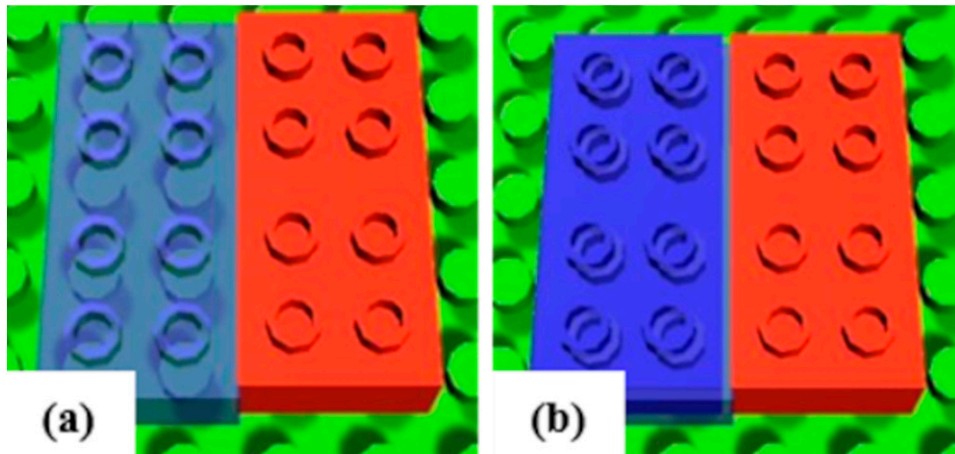

**Figure 1.** Visual realism in the DSTE: (**a**) the blue LEGO brick, which simulates the real one, is rendered in "opaque" mode, whereas the red product model used as a visual asset is rendered in "transparent" mode; (**b**) the misalignment between the simulated real brick and the corresponding visual asset to reproduce the effect of registration error.

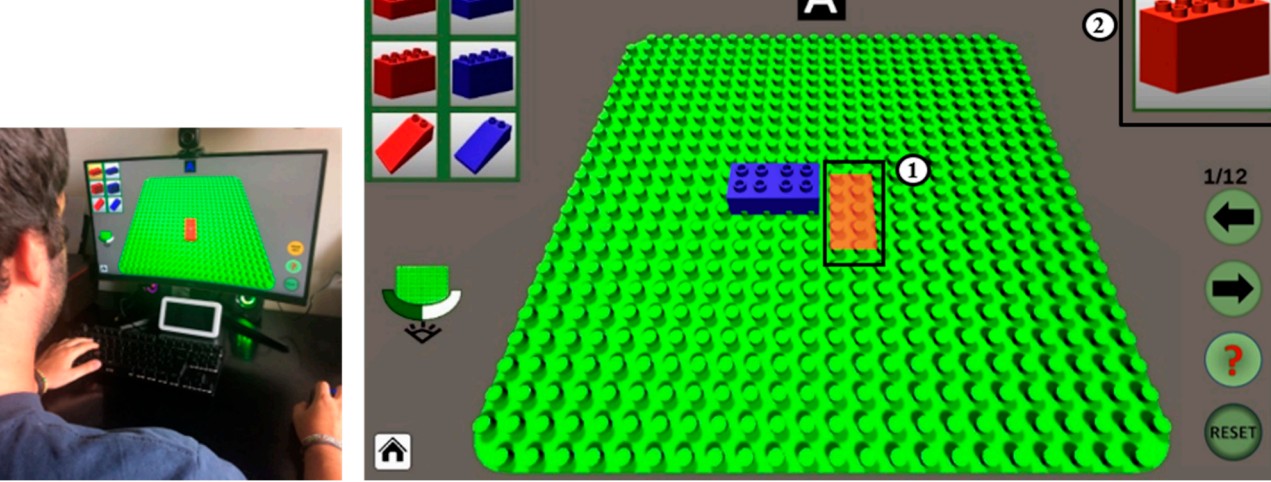

**Figure 2.** A user while carrying out the experiment with our DSTE at home (**left**); interface of the proposed DSTE (**right**): simulated AR instruction is provided through an auxiliary model used as a world-fixed visual asset ①, and a drawing used as a screen-fixed visual asset ②.

### 3.2. Three-Dimensional Interaction in the DSTE

To accomplish the assembly task, users interact with LEGO bricks through the following actions:

- Picking the assigned brick;
- Rotating a piece before putting it in place;
- Assembling a brick on the plate or on other pieces;
- Disassembling a brick (e.g., wrongly placed);
- Changing the point of view.

For each of these actions, we defined interaction metaphors in the DSTE following other UIs (e.g., Virtual Reality testing applications [35] and videogame GUIs) so that they would appear familiar to users.

**Picking.** Users can pick a single brick to place on the plate by clicking on the corresponding picture in the top-left corner of the interface, as in [35] (see Figure 3a), and then a 3D preview of the LEGO brick chosen appears on the GUI in the position of the mouse cursor (see Figure 3b). We decided to simplify this interaction from its homologous counterpart in true AR by not inserting the 3D models of the picking bins in the virtual environment. The main reason for this choice is that our testing environment is not aimed at evaluating the picking task. For the same reason, we showed only pictures of the bricks that are used in the LEGO set. As shown in other works [40,41], the proposed interaction metaphor allows users to focus their attention only on the assembly task and on the different types of instructions without the risk of decreasing their performance due to the picking task.

**Rotating**. When users pick a LEGO piece, they can also rotate it, clockwise or counterclockwise, by pressing the keys "D" and "A" on the keyboard, respectively (see Figure 4). Every time users press these keyboard buttons the picked brick rotates 90 degrees in the corresponding orientation. The change in angle takes place instantaneously without animations. No other rotation axes were considered because the assembly can be performed only along the vertical direction.

**Assembling**. When users understand where to place the LEGO brick, they can release it by clicking on the desired location. The application has been designed in such a way that it allows users to automatically attach the picked LEGO brick to the LEGO Duplo plate or to another placed brick, avoiding unintentional interpenetration. Our assembly action is intended to maintain naturalness and intuitiveness, but it is also simpler and faster than assembly actions proposed in other works [13–16], in which complex devices were used

to simulate real gestures. In this way, it is possible to avoid technical issues in the software physical simulation, such as the unpredictability of virtual object behavior, collision, and attaching [11].

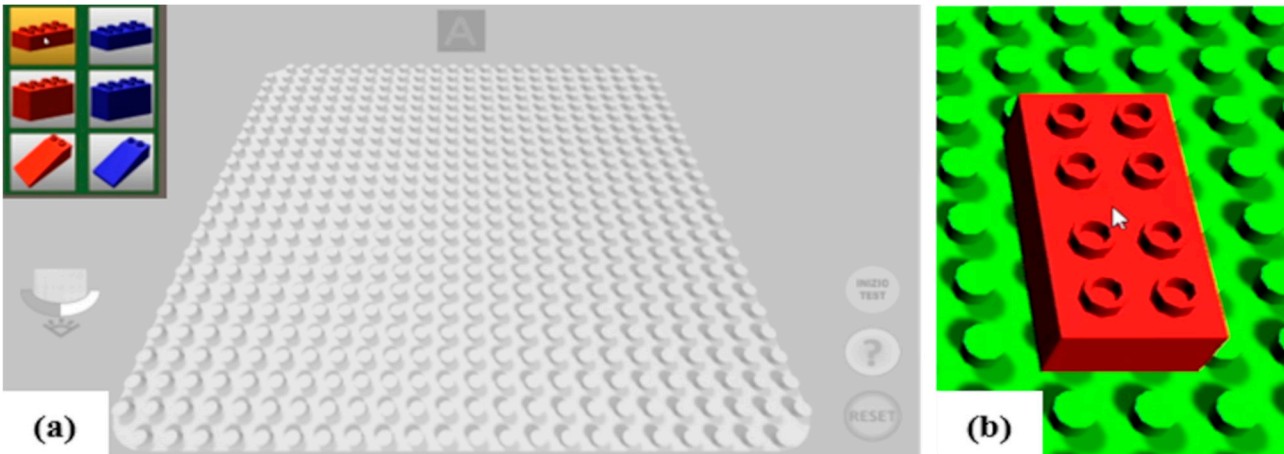

**Figure 3.** Picking metaphor: (**a**) the user picks a single LEGO brick to place on the plate by clicking on the appropriate GUI button; (**b**) a 3D preview of the real LEGO brick chosen appears on the GUI in the position of the mouse cursor.

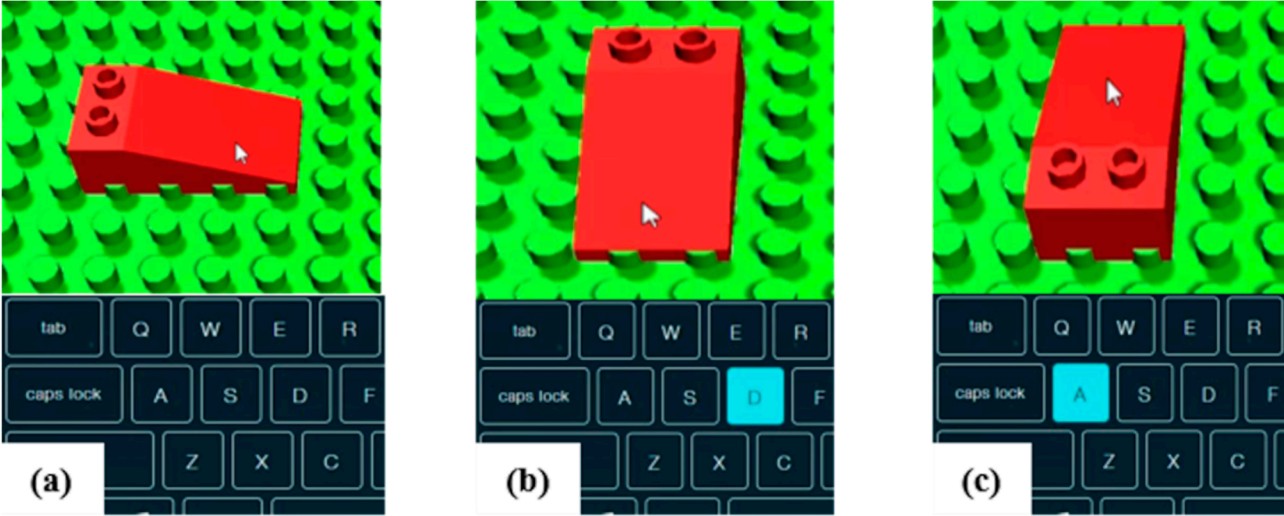

**Figure 4.** Rotating metaphor: (**a**) initial orientation and that obtained with (**b**) a clockwise rotation, (**c**) or a counterclockwise rotation of the picked LEGO brick.

**Disassembling**. If users make a mistake, an "undo" button allows them to remove the last piece that they wrongly placed. The "undo" button is placed on the top of the interface, and it allows the user to instantly remove the LEGO brick that needs to be disassembled (see Figure 5). Additionally, this interaction metaphor is much more simplified than its homologous counterpart in true AR, neglecting all physical aspects that do not concern our study.

**Changing the point of view**. A slider on the GUI allows participants to rotate the camera to change the point of view (see Figure 6). The action of the slider simulates the relative rotation between the user's head and the LEGO plate in true AR. Users have to hold down the left mouse button on the slider and move the cursor in the direction in which they want to change the view. In the editing phase, the test designer can choose the most suitable parameters for natural usability, such as the rotation speed of view and the main camera field of view and orientation.

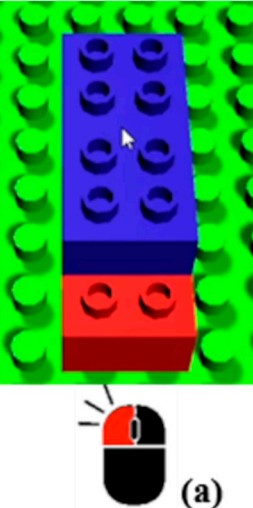 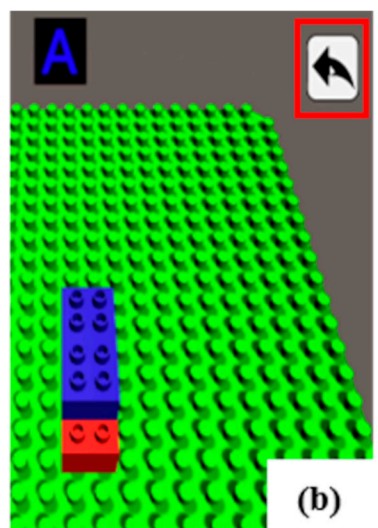 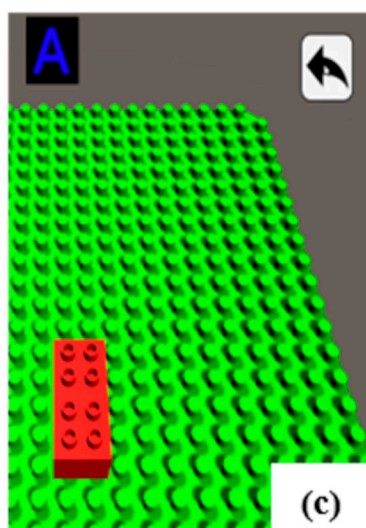

**Figure 5.** Assembling and disassembling metaphors: (**a**) the picked blue LEGO brick is attached to the placed red brick in a way different from that required by the instruction; (**b**) an "undo" button allows the user to delete the last blue LEGO brick that they wrongly placed; (**c**) the brick that needs to be disassembled is instantly removed after clicking on the "undo" button.

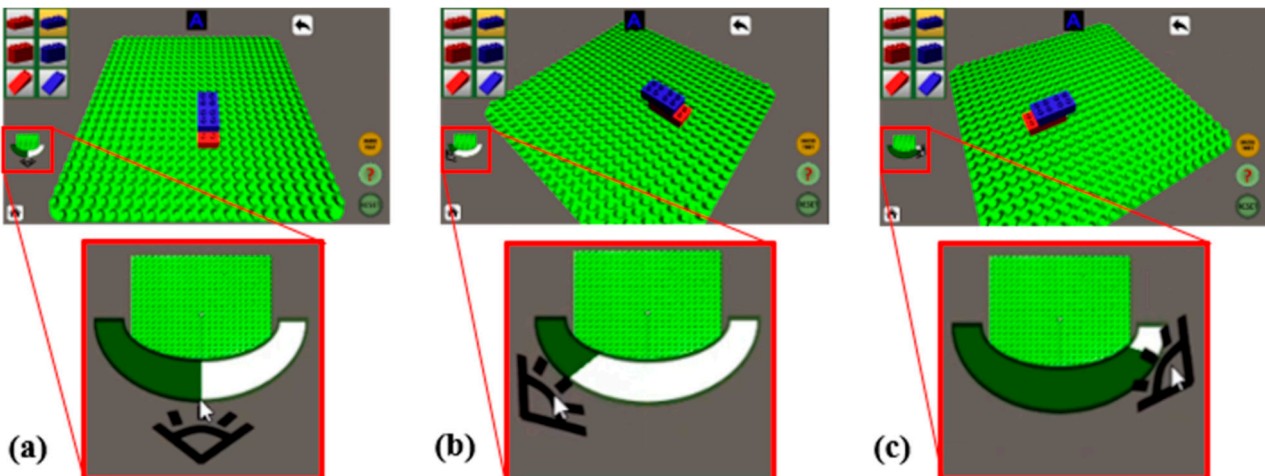

**Figure 6.** Slider used to change the camera point of view: (**a**) default camera orientation, (**b**) left rotation of the camera point of view, and (**c**) right rotation of the camera point of view.

### 3.3. DSTE Description

**Implementation.** We developed the DSTE on Unity 3D Engine. LEGO 3D CAD models were imported in Unity 3D, while all of the simulated AR information needed to be modeled in the software environment. Unity 3D is used to create the virtual scenes and set all experimental parameters (editor mode). Considering that no resolution limitation has been imposed on users' computer screens, in Unity, the GUI Canvas Scaler was set in "Scale with screen size" mode to ensure the correct anchoring of the GUI buttons. Then, the proposed DSTE is built into an executable file, which needs to be sent to the participants (build mode). It was successfully tested with Windows and macOS operating systems. During the experiment, the application acquires and stores (in a simple ASCII text file) the time data for each testing condition. The output file is then sent back to the experimenters.

**GUI buttons.** During the experiment, users are asked to click a button to start. Then, they can execute the assembly task, selecting and placing the assigned LEGO pieces. When they place a LEGO brick, they can continue with the following step by clicking a button (right arrow in Figure 2) in the GUI. Users can also move to the previous step in case

of a mistake through another button (left arrow in Figure 2). The completion time is measured for each trial. It is automatically stored in the output ASCII text file. If users are disturbed during the trial, they can move away from their computer and come back by clicking a button to restart that trial ("reset" button in Figure 2). All of these buttons are in the bottom-right corner of the GUI. In this current version of the DSTE, errors are not automatically detected. Thus, if experimenters need this information, they must either watch the recordings of the experiment or follow the users while doing it and manually annotate errors.

**Dual-task exercise.** The DSTE can assess the mental workload in the assembly task using a dual-task paradigm, inspired by Brunken's work [42]. At various time intervals, users receive a signal in the form of the color change of a letter placed at the top center of the interface (letter "A" in Figure 2). They have to quickly respond to this signal by pushing the "S" key on the keyboard. After this action, the color of the letter reverses again, and the software records and stores the reaction time in the ASCII output file. Then, after a time interval that can randomly range from 8 s to 18 s, there is another signal, and so on. In the editing phase, the test designer can change the time interval within which users have to respond to the dual-task exercise.

**Training scene.** We inserted a training scene in the testing environment to help users become familiar with all possible interactions. When users start the training, a popup window is displayed. It contains short text, images, and videos explaining how users should interact with the application. It can also be recalled during the execution of the experiment through a GUI button (question mark in Figure 2). During the training, they can build abstract shapes following (or not) the instructions provided. The verification of the training progress is left to users or, if present, to the experimenter who follows users remotely.

**Assembly trials.** After the training, users start the assembly trials for the experiment. It is possible to create customized trials according to the task complexity and the visual assets that must be tested. The designer can create various "LEGO sets" corresponding to different combinations of LEGO bricks used in the assembly to create various task complexities. Then, for each LEGO set, it is possible to test various combinations of visual assets through different target shapes.

## 4. Evaluation of the DSTE

### 4.1. Pilot User Study

The proposed DSTE was applied in a user study, where the goal was to determine the minimum amount of information that must be conveyed through AR in assembly tasks with different degrees of task complexity. We created four LEGO sets that need an incremental amount of information required because of the increasing number of types (Brick 2 × 4, Brick 2 × 4 × 2, Slope Brick 45 2 × 4) and colors (red and blue) of used LEGO bricks. We tested four combinations of visual assets: (i) an auxiliary model without color coding, (ii) an auxiliary model with color coding, (iii) a drawing and auxiliary model with color coding, and (iv) an animated product model with color coding.

We assessed a focus group with 6 people (1 female, 23 to 35 years old, mean = 29.5, SD = 4.85) on our staff who were involved as experimenters. They followed a balanced number of participants and contributed to the analysis of the results. In this way, they could provide important feedback on the effectiveness of the DSTE for this kind of experiment.

### 4.2. Procedure

The entire experiment carried out for each user was divided into two parts: the pilot user study, where they used the DSTE, and a subsequent subjective questionnaire. Participants carried out the experiment using their personal computers in their homes under our remote collaborators' supervision. At the end of the pilot study, users were asked to fill in a subjective questionnaire. Participants evaluated their user experience in the use of the DSTE through the UEQ. Then, they were asked to estimate their satisfaction

level with the DSTE regarding the five interaction metaphors with the simulated LEGO bricks. They also estimated the interactions they would have with a true AR application manipulating real LEGO bricks, which was shown through a video.

### *4.3. Participants*

We recruited 40 people (31 males, 12 to 49 years old, mean = 24.6, SD = 5.08) from local college students and staff and the authors' relatives. They were the same for the pilot study and the evaluation of the testing environment. We checked that no users were colorblind and that they were familiar with LEGO bricks, rating 4.1 on average (SD = 0.93, Median = 4, Max = 5, Min = 2) on a 5-point Likert scale (1: Not at all familiar; 5: Extremely familiar). As regards familiarity with AR applications, the rating was 2.7 on average (SD = 1.48, Median = 3, Max = 5, Min = 1). The average completion time of the test was 50 min (40 for the pilot study and 10 for the questionnaire). The Microsoft Teams platform was used by the experimenters to follow each participant and annotate errors.

### *4.4. Results*

In terms of the reported user experience with the DSTE, the average scores of the six measures captured by the UEQ were 1.88 (SD = 0.74) for attractiveness, 1.98 (SD = 0.90) for perspicuity, 1.67 (SD = 0.98) for efficiency, 1.56 (SD = 0.71) for dependability, 1.74 (SD = 0.89) for stimulation, and 1.54 (SD = 1.00) for novelty. Figure 7 contextualizes these scores into the global UEQ benchmark [22].

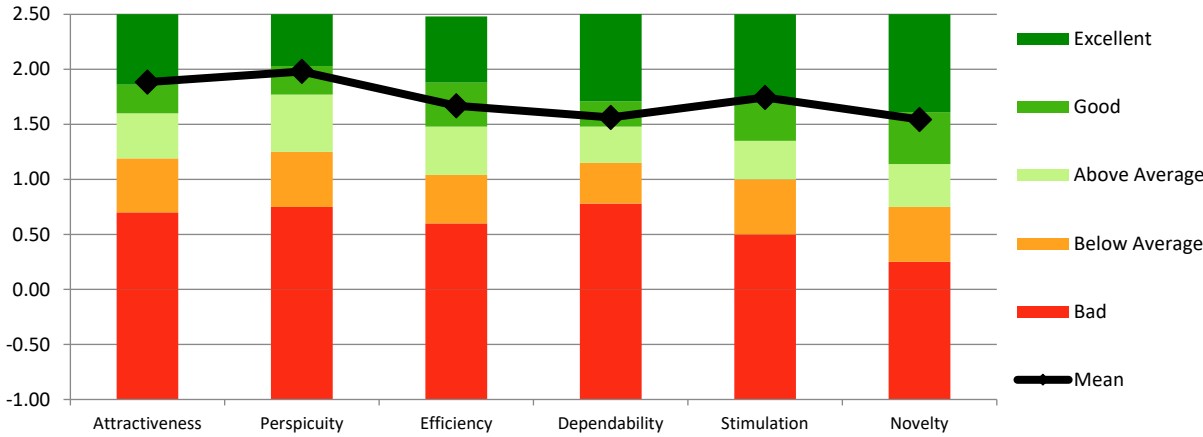

**Figure 7.** The User Experience Questionnaire (UEQ) scores reported by participants compared to the UEQ benchmark [22].

We used the Wilcoxon rank-sum test to compare the satisfaction between the proposed interaction metaphor and the real interaction with true AR (see Figure 8). For three out of five interactions, there were no statistically significant differences (Z = −0.041, *p* = 0.967 for "picking"; Z = −0.892, *p* = 0.372 for "assembling"; Z = −1.346, *p* = 0.178 for "rotating"). There is statistically significantly higher satisfaction for the "disassembling" metaphor (Z = −2.812, *p* = 0.005) compared to disassembling real LEGO bricks. We observed a statistically significantly lower satisfaction for the "changing point of view" metaphor (Z = −2.297, *p* = 0.022) compared to rotating the head (or the plate) with true AR.

The results of the pilot user study are out of the scope of this work. However, it is important to note the main feedback derived from the focus group:

- The data gathered with the DSTE were useful for answering the research questions of the pilot user study.
- With the DSTE, it is possible to plan experiments in a more flexible way.
- None of the pilot study participants opened the training popup window during the trials, only during the initial training.

- Participants often forgot to perform the secondary task, and the experimenters had to remind them.
- If errors were automatically recorded, it would be possible to exclude the experimenter following live participants.
- None of the participants felt tired; rather, they were focused and excited throughout the experiment.

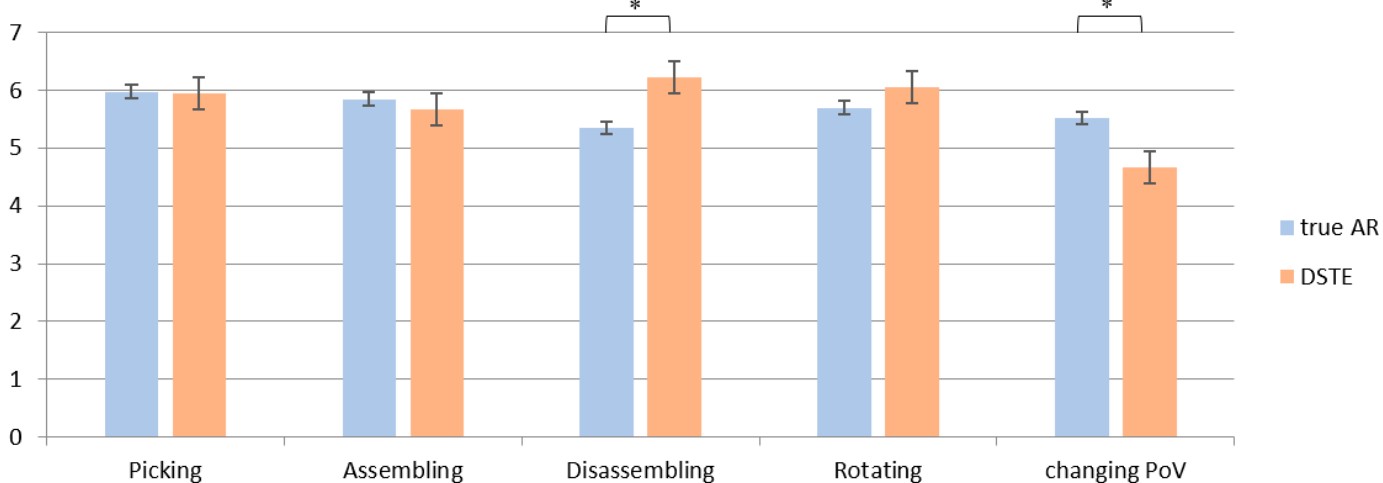

**Figure 8.** Mean values of user satisfaction reported by participants for interaction with LEGO bricks in the DSTE and in true AR. The asterisks indicate statistically significant different conditions.

## 5. Discussion

The user evaluation of the proposed DSTE was useful for an initial answer to our research questions. We applied it in a pilot user study, where we looked for visual assets that convey the needed information without redundancy. The DSTE was useful for the research goals of that user study, as reported by the focus group and further confirmed by the results of the UEQ. Users judged the pragmatic qualities of efficiency, perspicuity, and dependability as "good". Additionally, the hedonic quality of novelty can be considered "good", while attractiveness and stimulation are "excellent". This result confirms what was reported by the experimenters, who observed that participants were focused and excited during the experiment, even though it lasted 50 min on average. This outcome was also achieved due to the intuitive interface and the fact that users conducted the experiments from home. These factors themselves reinforce the great potential offered by the emerging Metaverse concept. It turns out to effectively be a paradigm in which technologies and the contents of AR and VR are expanding and evolving into the virtual world, connecting people together regardless of their location. However, some improvements are needed to the DSTE regarding the dual-task exercise and error counting to allow users to perform the experiments autonomously, without invalidating the results. An alternative secondary task (e.g., a sound) or an automatic reminder could be evaluated in future works.

The results about user satisfaction confirm the appropriateness of our choices regarding the proposed interaction with simulated LEGO bricks. Only the slider used to rotate the point of view needs to be reconsidered. Its lower satisfaction may be related to the rotation speed, but it can be easily reduced in the editing phase. Another possible reason is that changing the point of view with the slider requires a task performed by the hands, while in true AR, it can be obtained by normal head movement. All other interactions require a task performed by the hands both in the DSTE and in true AR. The better satisfaction with the "disassembly" interaction metaphor may be attributable to the fact that it is the easiest task, requiring pushing a button rather than detaching a LEGO brick, which may compromise the entire LEGO assembly due to unwanted movements.

The results obtained regarding user experience and interaction metaphors allowed us to validate our DSTE as a framework to conduct experiments on information presentation

in AR. Our proposal was implemented with Unity 3D. The interface could also have been realized with other simulation software, such as Blender and Maya. However, doing so would require integrating additional software to develop an application capable of ensuring the right metaphors for human–computer interaction or collecting data automatically. On the other hand, Unity 3D is standalone software for creating an application to conduct experiments remotely, especially for our framework. Moreover, if there is a need to move to true AR, developers can easily make a porting of the application considering that Unity 3D is seamlessly integrated with Vuforia Engine as an AR platform.

In the presented DSTE, we decided to use assembly tasks based on LEGO Duplo bricks because they are an established method to minimize the effect of user experience and to generalize tasks [17–19]. The aim was to propose an assembly task (or a disassembly one in case of a wrongly placed brick) to reproduce behavior similar to true AR without worrying about additional factors unrelated to the experiment, such as optimizing tracking. Indeed, an alternative solution to reach this goal consists of distributing a mobile AR application to users on their smartphones/tablets [37]. Therefore, in the case of an experiment with true AR conducted in this way, a fiducial marker could be an example of a tracking method that can be used as a reference system for AR information on the LEGO Duplo plate. After establishing the right location, further testing would have to be carried out in the development phase to ensure optimal visual asset registration based on the LEGO bricks to be put in place, resulting in wasted time. Moreover, experimenters should also provide users with all LEGO pieces used in the experiments. However, the main limitation of this solution, compared to ours, is that users would carry out experiments in an uncontrolled environment as regards, e.g., lighting, the relative position of the camera, and LEGO bricks, with the risk of invalidating the results of the study. A second alternative method, already described in the literature, is recording a video prototype illustrating the AR application and gathering data from users through online surveys [43–45]. However, participants' responses might not reflect their opinions on actually using the interface [43]. Furthermore, it is not possible to ask users to accomplish tasks using the AR application; thus, only subjective data can be collected.

The proposed DSTE is aimed only at user studies on information presentation in AR. It can help to test different combinations of visual assets and their properties (e.g., world-fixed/screen-fixed, shading, size, and animation) involving a wide range of participants. Other studies on AR (e.g., interaction, display comparison, and perception) are more affected by real testing conditions. Thus, a tailored simulated testing environment is needed.

Nevertheless, our solution does not lack limitations. In this work, we have demonstrated that our DSTE is valid for replacing laboratory experiments with remote ones with controlled variables (lights, camera position, and LEGO pieces). Still, there may be limitations in more in-depth studies on how AR information can be visualized for specific hardware and real industrial constraints. Moreover, despite our efforts to achieve the desired level of visual realism, a user unfamiliar with AR might lose the perception between true and simulated AR, leading to invalid results. This ambiguity could also occur in understanding how true AR works through the proposed video. However, our sample of users had average knowledge of AR, so these problems were not encountered in our user study.

## 6. Conclusions

In this work, we propose using a desktop-simulated testing environment (DSTE) to carry out experiments on information presentation with AR remotely. It exploits simulated AR as a precursor of the emerging concept of the Metaverse with the aim of connecting people together, regardless of their location. Moreover, contrary to similar prototypes already present in the literature, the interface does not need complex devices to be used. A user's personal computer equipped with a mouse and keyboard is enough, thus facilitating data acquisition from many more users. The DSTE allows accomplishing assembly tasks



on LEGO Duplo bricks with different task complexities. The assembly instructions are provided through the AR visual assets that must be analyzed in the user study. We successfully applied the DSTE in a pilot user study and obtained promising results for both the overall user experience and user satisfaction with the proposed interface design. These results, confirmed by the experimenters' feedback, are important for understanding how to improve the DSTE in future works, making it easily scalable to other similar experiments.

**Supplementary Materials:** The following supporting information can be downloaded at: https://www.mdpi.com/article/10.3390/app122412600/s1. Table S1: UEQ and User Satisfaction data; Video S1: DSTE interface and usage.

**Author Contributions:** Conceptualization, Methodology, Formal Analysis, Investigation, and Writing—Original, M.G.; Methodology, Software, Visualization, Investigation, and Writing—Original Draft, E.L.; Software, Investigation, and Writing—Review and Editing, A.E.; Resources, Funding Acquisition, and Writing—Review and Editing, M.F.; Supervision, Resources, Funding Acquisition, and Writing—Review and Editing, A.E.U. All authors have read and agreed to the published version of the manuscript.

**Funding:** This research was funded by the Italian Ministry of Education, University and Research under the Program "Department of Excellence" Law 232/2016, grant number CUP-D94I18000260001.

**Informed Consent Statement:** Informed consent was obtained from all subjects involved in the study.

**Data Availability Statement:** Data are provided in Supplementary Materials.

**Acknowledgments:** The authors would like to acknowledge all participants involved in the experiment.

**Conflicts of Interest:** The authors declare no conflict of interest.

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
