# Peer review of "Towards the Evaluation of Augmented Reality in the Metaverse: Information Presentation Modes"

_applsci, doi:10.3390/app122412600_

Round 1
Reviewer 1 Report
The proposed work is interesting, specifically focused on User studies of AR- Information
Presentation in the Metaverse. Although the work is precise and could make a valuable
contribution, I believe that it needs to be updated to increase its applicability and make it
ready for publication.
In the abstract section, the complexity of user studies is explained in general and not
clear in terms of the stated proposed work.
Eliminate the Redundancy of content
Still numerous challenges exist in AR for its practical implementation at the industry level for various applications. Please refer some recent literature and include. The following keywords would help further (Challenges and opportunities on AR/VR technologies; Augmented reality-based guidance;)
Please state the motivation of the Work with reference to the state of the art.
How the current framework is better than the existing simulation software (like blender and Maya virtual software)
Explain in terms of applicability for any to justify the completeness of the framework.
(Fig.5) Most of the tasks explained from the assembly/disassembly perspective please discuss on object identification and tracking challenges.
It is a very good work, Integration of assembly task planners to handle the uncertainties in the working environment can improve the quality of manuscript further. (optional to consider or can be included a future scope).Please refer
Wang, Z. B., et al. "Assembly planning and evaluation in an augmented reality environment." International Journal of Production Research 51.23-24 (2013): 7388-7404.
Bahubalendruni, MVA Raju, et al. "A hybrid conjugated method for assembly sequence generation and explode view generation." Assembly Automation (2019).
Reinhart, G., and C. Patron. "Integrating augmented reality in the assembly domain-fundamentals, benefits and applications." CIRP Annals 52.1 (2003): 5-8.
Reviewer 2 Report
This paper presents a DSTE to conduct experiments on information presentation with AR also from remote. The authors conduct a study to verify their technique's usability and compare it to true AR technique.
The paper is clearly written and structured, although a careful editing pass is required(like the figure size and format).
In reading the introduction, I felt that the motivation and gap addressed could be conveyed more clearly. I agree that the DSTE would be important in Metaverse, but the experiment only seems like how to implement a LEGO program. And in the Related work part, I understand because the VR device is expensive, so this paper proposed a desktop-based application with a mouse and keyboard. However, the mouse and keyboard are too usual devices, and there are challenging to do some new tech. There would have some better choices, like webcam(which is also cheap) to recognize the hand to conduct the experiment.
And the comparison experiment conducted with True AR needs more details and results, such as pictures. The true AR evaluated in the form of video without personal experience, it is difficult to verify the conclusion by comparing the experimental results. Given the ample lego program and literature around alternative techniques, the submission's contribution is too small without a direct comparison with true AR through a user study.
The paper isn't in a state suitable for acceptance at this time. The experiment is weak, and I am not convinced that the technical contribution is sufficient for journal acceptance. Therefore, I am currently on the side of rejection.
Reviewer 3 Report
Congratulation for this very interesting work!
My remarks:
- Authors should explain what UEQ is (when first mentioned) and give some details / citations.
- Authors should clearly describe the (potential) limitations of their study.
- Please reformulate this phrase "The intuitive interface and users' being at home helped in obtaining this result strengthen the great potential offered by the emerging concept of Metaverse."
- Very little space is dedicated to Results and Discussion compared to the rest of the paper. Is the intention of the authors to leave other results for upcoming papers?
Small formatting issues:
- Subchapters in section 4 are all numbered as 4.1.
- The "main feedbacks derived from the focus group" are improperly formatted
Round 2
Reviewer 2 Report
The discussion part of the article has improved.